# Pure Solid Pattern of Non-Small Cell Lung Cancer and Clustered Circulating Tumor Cells

**DOI:** 10.3390/cancers14184514

**Published:** 2022-09-17

**Authors:** Noriyoshi Sawabata, Takeshi Kawaguchi, Takashi Watanabe, Daiki Yohikawa, Noriko Ouji-Sageshima, Toshihiro Ito

**Affiliations:** 1Department of Thoracic and Cardio-Vascular Surgery, Nara Medical University, Kashihara 634-8552, Japan; 2Department of Immunology, Nara Medical University, Kashihara 634-8521, Japan

**Keywords:** non-small cell lung cancer, pure solid, computed tomography, clustered circulating tumor cell, prognosis

## Abstract

**Simple Summary:**

There are two solid patterns of non-small cell lung cancer (NSCLC) on computed tomography (CT): pure or mixed with ground glass opacities (GGOs). They predict the degree of tumor invasiveness, which may suggest the presence of clustered circulating tumor cells (CTCs), a predictor of poor prognosis. We assessed the implications of the solid patterns on CT and the preoperative clustered CTCs. Pure solid appearance was an independent predictor of preoperative clustered CTCs in the multivariable analysis, and preoperative clustered CTCs were an independent predictor of poor recurrence-free survival; the solid pattern was not an independent variable.

**Abstract:**

There are two solid patterns of non-small cell lung cancer (NSCLC) on computed tomography (CT): pure or mixed with ground-glass opacities (GGOs). They predict the degree of invasiveness, which may suggest the presence of clustered circulating tumor cells (CTCs), a predictor of poor prognosis. In this study, we assessed the implications of the solid patterns on CT and the preoperative clustered CTCs in surgically resected NSCLC. CTCs were detected using a size selection method. The correlation between the presence of preoperative clustered CTCs and the solid pattern and the prognostic implications were evaluated using co-variables from the clinical-pathological findings. Of the 142 cases, pure solid lesions (Group PS) and mixed GGOs (Group G) were observed in 92 (64.8%) and 50 (35.2%) patients, respectively. In Groups PS and G, clustered CTCs were detected in 29 (31.5%) and 1 (2.0%) patient (*p* < 0.01), respectively. The PS appearance was an independent predictor of preoperative clustered CTCs in the multivariable analysis, and preoperative clustered CTCs were an independent predictor of poor recurrence-free survival; the solid pattern was not an independent variable. Thus, the PS pattern of NSCLC on CT is an indicator of preoperative clustered CTCs, which is an independent poor prognosis predictor.

## 1. Introduction

Lung cancer is one of the leading causes of cancer-related deaths worldwide. Despite the advances in diagnostic strategies and treatment, it has a poor prognosis [1]. The prognosis of surgical cases of non-small cell lung cancer (NSCLC) depends on the histological diagnosis [2]. Unlike non-lepidic lesions of NSCLC, lepidic lesions are histologically non-invasive and commonly observed as ground-glass opacities (GGOs) on computed tomography (CT) [3]. Mixed-type GGOs (presence of both GGOs and a solid region) on CT in patients with surgically resected lung cancer are associated with a better survival than pure solid lesions [4,5,6].

A solid appearance on CT indicates a high density of remodeled pulmonary tissue, such as stromal reaction in adenocarcinoma or invasive lung cancer of any histological type [7,8]. The probability of detecting cancer-associated fibroblasts (CAFs), tumor-associated macrophages (TAMs), and hypoxia-related factors is reportedly lower in part-solid lesions of lung cancer than in the pure solid types [9], where the tumor environment has the potential to induce cancer stem cells [10] and epithelial–mesenchymal transition (EMT) [11]. Cancer stem cells and EMT have the same features with clustered circulating tumor cells (CTCs), which are precursors of metastasis [12,13]. Furthermore, clustered CTCs predict postoperative early recurrence [14]. 

Currently, there are no in-depth studies on whether the solid pattern on CT is a predictor of clustered CTCs, which indicate a poor prognosis [15]. Therefore, in this study, we aimed to evaluate the implications of a solid pattern on CT and the presence of preoperative clustered CTCs in surgically resected NSCLC.

## 2. Materials and Methods

### 2.1. Ethical Considerations

The study protocol was approved by the Institutional Review Board of the Hoshigaoka Medical Center (No. 1412; 15 April 2014) and Nara Medical University Hospital (No. 1718; 21 January 2018). Informed consent was obtained from all the study participants.

### 2.2. Evaluation Outcomes

The study outcome was the presence of clustered CTCs in peripheral arterial blood extracted preoperatively in patients with NSCLC according to the solid pattern on CT (concomitant with GGOs or pure solid). The other study outcomes included recurrence-free survival (RFS) and overall survival (OS). 

### 2.3. Settings and Patient Selection 

Of the patients who underwent CTC evaluation preoperatively for pulmonary nodules at either the Nara Medical University Hospital or Hoshigaoka Medical Center between April 2014 and October 2021, those with NSCLC were included and examined. The inclusion criteria were (1) patients with pulmonary nodules observed on CT, (2) patients who underwent preoperative CTC testing, and (3) patients that had a confirmed pathological diagnosis of NSCLC. The exclusion criteria were (1) pathological diagnosis of non-NSCLC and (2) no provision of consent.

### 2.4. Definition of Pure Solid Lesions and Non-Pure Solid Lesions on CT

Regarding the morphological findings on CT, the CT (pulmonary field condition, 1-mm sliced axial cross section) findings were registered in the patients’ medical records by a radiologist. The state of the substantial component containing GGO lesion, overall tumor diameter, and solid component diameter were selected.

### 2.5. Clinicopathological Parameters

The clinicopathological characteristics included demographics, tumor size (whole size on CT, size of the solid lesion on CT, and gross size), tumor lesion appearance on CT, tumor markers (carcinoembryonic antigen [CEA], and cytokeratin 19 fragment [CYFRA]), and maximum standardized uptake value (SUVmax) in the tumor as shown using the Discovery^®^ series of devices (GE 121 Healthcare, Little Chalfont, UK) in accordance with fluorodeoxyglucose-positron emission tomography and positron emission tomography/CT medical guidelines [16]. The pathological characteristics of tumor lesions included histological diagnosis (none/minimally invasive adenocarcinoma or not, and involvement of lepidic lesions), and the grade of invasiveness of the tumor lymph duct, tumor vessel, and pleura. Spread through air space (STAS) was assessed using pathology reports. Additionally, the pulmonary resection method was chosen as a parameter for analysis. As a rule, mediastinal lymph node removal was performed systematically for lobectomy or pneumonectomy cases but was case-dependent for segmentectomy or wedge resection cases.

### 2.6. Detection of CTCs

Peripheral arterial blood (3 mL) was collected in ethylenediaminetetraacetic acid tubes preoperatively. Tumor cells in the blood samples were extracted with a CTC selection kit using a size selection method (ScreenCell^®^ CYTO Kit, ScreenCell, Paris, France). The extracted cells were stained using the hematoxylin–eosin method and observed under a light microscope, using a previously published method [17].

CTCs were diagnosed by referring to an atlas of cytology on CTCs from solid cancers [18]. If suspected CTCs were detected, they were not diagnosed as CTCs. CTC detection was classified into three categories: no tumor cells detected (ND), single cells detected (S), and clustered cells (≥4 tumor cells) detected (C). Gathering tumor cells containing two or three cells were not considered as one cluster but as two or three single cells (Figure 1). To count the total number of CTCs, a lump of clustered CTCs was counted as one.

The variables for the logistic regression analysis of preoperative cluster CTC detection included invasive size > 2 cm, tumor vessel invasion [V(+)], presence of STAS, serum CEA level > 7.5 µg/dL, CYFRA > 3.3 ng/mL, SUVmax > 2.9, pulmonary sublobar resection (wedge resection or segmentectomy), and pathological stage III or V, in addition to solid patterns on CT.

### 2.7. Prognosis Evaluation

The parameters of the prognosis assessment included sex difference, pure solid patterns on CT, status of clustered preoperative CTCs, invasive tumor size > 2.0 cm, V(+), STAS, CEA level > 7.5 μg/mL, CYFRA > 3.3 ng/mL, SUVmax > 2.9, pulmonary resection method, p stage III, p stage IV, and adjuvant therapy. Hazard ratios were calculated in univariate and multivariate analyses to evaluate the independency of the solid CT pattern and clustered CTCs. All the patients were followed-up at 1–3-month intervals during which physical examination, chest radiography, and testing for blood tumor markers were performed. Thoracoabdominal CT was also performed every 6 months. The mortality and recurrence data were collected by a primary physician (NS). The median follow-up period was 52 (range, 3–72) months, and the last follow-up was in May 2022. Furthermore, the survival curves of the group bisected by solid patterns on CT or the presence of preoperative clustered CTCs were drawn. 

### 2.8. Statistical Analyses

Statistical analyses were performed using the freely available statistics software “EZR” (Easy R), which is based on R and R commander [19].

Fisher’s exact test was used to compare two dichotomous variables, and the *t*-test was used to compare the mean values. The chi-square or Kruskal–Wallis test was used for comparisons between multiple groups. A logistic regression analysis for detecting CTC clusters was performed using selected clinicopathological parameters as covariables. Regarding prognosis analyses, RFS and OS rates were calculated. Survival curves were obtained using the Kaplan–Meier method and assessed using the log-rank test. Hazard ratios were calculated using the Cox proportional model in the univariate and multivariate analyses. Only significant variables from the univariate analysis were included into the multivariate analysis. The level of statistical significance was set at *p* < 0.05.

## 3. Results

### 3.1. Characteristics of the Patients

Of the 179 patients with pulmonary nodules who underwent preoperative testing to detect CTCs at Nara Medical University Hospital (*n* = 78) or Hoshigaoka Medical Center (*n* = 101), 37 were not diagnosed with NSCLC on pathological examination (metastasis, *n* = 33; non-specific reactive inflammation, *n* = 4). Therefore, 142 patients whose characteristics are shown in Table 1 were included in this study (Figure 2).

There were 60 (42.3%) females with a mean age of 69.4 years. Regarding the CT findings, pure solid lesions were detected in 92 (64.8%) patients, with a mean solid size of 2.3 cm. All the patients had either clinical stage I or II. Thirteen (9.2%) patients had advanced pathological stage (III or IV) tumors. Pulmonary resection was performed using wedge resection, segmentectomy, lobectomy, and pneumonectomy in 41 (28.9%), 13 (9.2%), 86 (60.6%), and 2 (1.4%) patients, respectively. Histologically, 17 (12.0%), 65 (45.8%), 78 (54.9%), 45 (32.6%), and 37 (26.1%) patients had non-invasive or minimally invasive adenocarcinoma, vessel invasion, lymphatic invasion, pleural invasion, and STAS, respectively. There was a significant difference in CT findings, clinical parameters, stage, operation, and histological findings between the two groups (Table 1).

### 3.2. CTC Status According to Solid Patterns on CT

The status of CTCs according to the solid patterns on CT is shown in Table 2. Clustered CTCs were detected in 30 (21.1%) patients: 29 (96.7%) in the pure solid group and one (0.1%) in the mixed GGO group (*p* < 0.01).

### 3.3. Predictor of Preoperative Clustered CTCs

The results of the logistic regression analysis for clustered CTC detection are shown in Table 3. In the multivariable analysis, pure solid patterns on CT and high CEA levels were positive predictors of preoperative clustered CTC detection. 

### 3.4. Survival Analyses

There were 34 cases of recurrence: local (surgical stump) recurrence, *n* = 2; regional (ipsilateral thorax) recurrence, *n* = 21; and distant metastases, *n* = 12. Thirty patients underwent adjuvant therapy. Concomitant recurrence was diagnosed in one patient (local plus regional recurrence at the surgical stump and ipsilateral mediastinal lymph node). Seventeen patients died from the original lung cancer (*n* = 11), other cancers (*n* = 2), cerebellar infarction (*n* = 1), acute respiratory distress syndrome (*n* = 1), or acute exacerbation of interstitial pneumonia (*n* = 1).

The survival curves for the patients with pure solid lesions on CT (Figure 3) and preoperative clustered CTCs are shown in Figure 4. The results of the Cox proportional hazards model analyses are shown in Table 4. Preoperative clustered CTC detection and pathological stage were independent predictors of RFS, unlike pure solid patterns on CT. Preoperative clustered CTC detection, high CEA levels, high CYFRA levels, and sublobar resection were significant predictors of OS in the univariate analysis. Moreover, sublobar resection was an independent predictor of OS in the multivariate analysis.

## 4. Discussion

In this study, clustered CTCs were rarely detected preoperatively in patients with NSCLC presenting with a mixed GGO pattern on CT. Furthermore, the presence of preoperative clustered CTCs was an independent prognostic predictor for RFS, unlike that for pure solid appearance on CT. Therefore, the worse prognosis of tumors with pure solid lesions on CT than tumors with mixed GGO may be explained by the presence of clustered CTCs, which is a poor prognostic indicator [15,20].

Cancer metastasis occurs predominantly via clustered CTCs from the original lesion [20,21]. In practice, detecting clustered CTCs during surgery for NSCLC is a predictor of early recurrence and poor prognosis [15,16,22]. This is because clustered CTCs have a high malignancy potential based on their EMT characteristics [12] and methylation, which is associated with transcription factors responsible for cancer stem cells [13]. These characteristics are present at the invasive site of the cancer, which appears as a non-lepidic region in lung cancer [9]. There are two types of non-lepidic regions (their presence is a surrogate sign of the invasive site of lung cancer) [23]: pure solid lesions and part-solid lesions with lepidic growth, where the malignancy potential is lower than that of a pure solid lesion. Furthermore, in the microenvironment of part-solid tumors, there is a lower chance of detecting hypoxia-induced factors (demethylation inducing), CAFs and TAMs [24], which are responsible for CTC clusters [25]. This can explain the significantly lower detection rate of preoperative cluster CTCs in the tumors with a non-pure solid appearance on CT with GGOs than in tumors with pure solid lesions, because GGO lesions are a surrogate marker of pathologically detected lepidic lesions. Moreover, this can also be explained by the significantly higher SUVmax in solid-lesion tumors than in lepidic growth-containing lesions, because the SUVmax is elevated in tumors with activated CAFs or TAMs [26]. 

Several recent studies have suggested that GGOs on CT are a predictor of better survival [14,15,16], even in hypermetabolic lung adenocarcinoma [27]. Since lepidic lesions in NSCLC appear as GGOs on CT [3], the affirmative prognosis of part-solid lesions in GGO cases might explain the tumor’s low potential to form clustered CTCs, which are a precursor of metastasis.

This study had several limitations. First, CTC detection requires inter-observer agreement between cytologists; therefore, pseudo-positive and pseudo-negative observations may have occurred. However, the specificity of this method is very high, as shown in our previous study [14,22] and a comparative method study [17]. Size selection method using a filter has high sensitivity to large cluster CTCs, so we think that it is suitable for research targeting cluster CTCs. Second, the small sample size from only two centers and the retrospective design may have introduced selection bias. Therefore, a large prospective study is required to confirm our findings. However, this is the first study to reveal that a pure solid appearance on CT may suggest clustered CTCs; in future, this concept may contribute greatly to the eradication of lung cancer recurrence.

## 5. Conclusions

Clustered CTCs, which are predictors of poor prognosis, were more frequently detected in cases of NSCLC with a pure solid appearance on CT than in tumors with mixed GGOs. This phenomenon explains the worse prognosis of patients with pure-solid tumors than those with mixed GGO tumors.

## Figures and Tables

**Figure 1 cancers-14-04514-f001:**
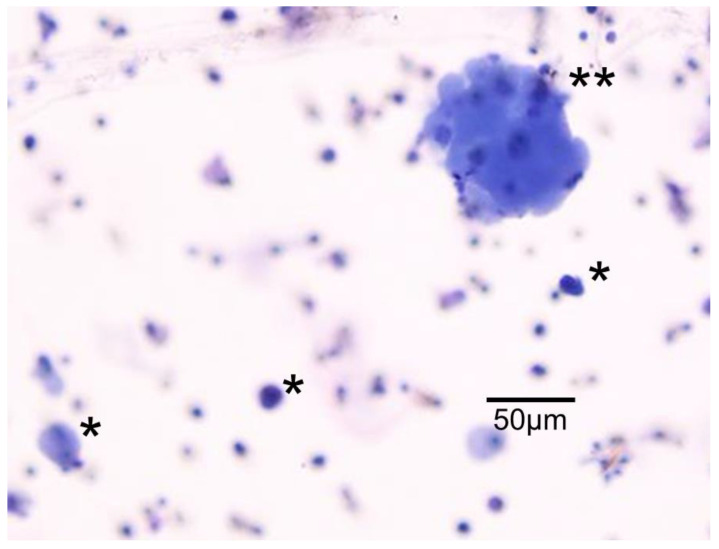
Circulating tumor cells (CTC) extracted with a CTC selection kit using a size selection method (ScreenCell^®^ CYTO Kit, Screen-Cell, Paris, France). *, singular CTC; **, clustered CTCs.

**Figure 2 cancers-14-04514-f002:**
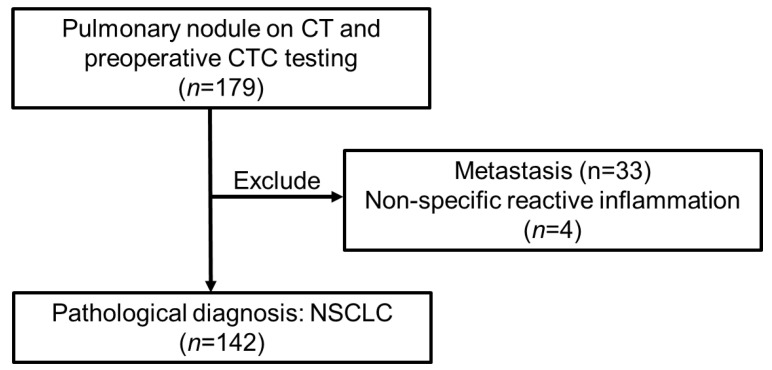
Patient selection flow chart. CT, computed tomography; CTC, circulating tumor cells; NSCLC, non-small cell lung cancer.

**Figure 3 cancers-14-04514-f003:**
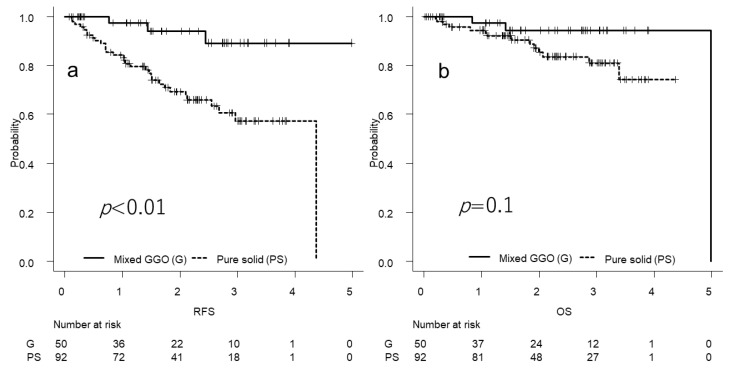
Survival curves according to the status of solid patterns on computed tomography. (**a**) The two-year recurrence-free survival (RFS) rate is 94.1% in the mixed GGO (G) group (*n* = 50) and 69.4% in the pure solid (PS) group (*n* = 92) (*p* < 0.01); (**b**) the two-year overall survival (OS) rate is 94.3% in the G group and 85.4% in the PS group (*p* = 0.1).

**Figure 4 cancers-14-04514-f004:**
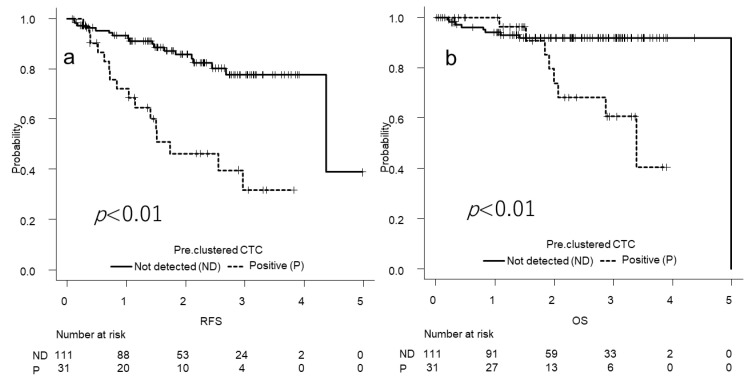
Survival curves according to the status of the circulating tumor cells. (**a**) The two-year RFS is 85.6% in the not detected (ND) cluster circulating tumor cell (CTC) (*n* = 111) group and 46.3% in the cluster CTC-positive (P) group (*n* = 31) (*p* < 0.01); (**b**) the two-year OS is 91.7% in the ND group and 71.5% in the P group (*p* < 0.01).

**Table 1 cancers-14-04514-t001:** Characteristics of the patients according to the solid patterns on computed tomography.

Variables			CT Findings		
		Total	Pure Solid	Mixed GGO	*p*-Value
N		142	92 (64.8%)	50 (35.2%)	
Demographics					
	Sex (Female)	60 (42.3%)	37 (40.2%)	23 (46.0%)	0.6
	Age, years	69.4 ± 8.8	68.8 ± 8.8	70.4 ± 8.8	0.3
CT					
	Whole size, cm	2.5 ± 1.5	2.7 ± 1.7	2.1 ± 0.9	0.02
	Solid size, cm	2.3 ± 1.7	2.7 ± 1.7	1.2 ± 1.1	<0.01
	Solid ratio, cm	0.8 ± 0.3	1.0 ± 0.0	0.5 ± 0.4	<0.01
Clinical parameters					
	CEA (µg/mL)	7.4 ± 12.9	8.6 ± 14.1	5.0 ± 10.0	0.1
	CEA > 7.5 (µg/mL)	32 (22.5%)	28 (30.4%)	4 (7.7%)	<0.01
	CYFRA (ng/mL)	2.4 ± 2.7	2.7 ± 3.2	1.9 ± 1.0	0.1
	CYFRA > 3.3 (ng/mL)	19 (13.3%)	16 (17.4%)	3 (6.0%)	0.1
	SUVmax	5.5 ± 5.9	7.4 ± 6.4	2.1 ± 2.2	<0.01
	SUVmax > 2.9	84 (59.2)	70 (76.1%)	14 (28.0%)	<0.01
Stage					
Clinical	I	118 (87.4%)	75 (81.5%)	50 (100%)	<0.01
	II	17 (13.3%)	17 (18.5%)	0 (0.0%)	
Pathological	I	109 (76.8%)	61 (66.3%)	48 (96.0%)	<0.01
	II	20 (14.1%)	18 (19.6%)	2 (4.0%)	
	III	7 (4.9%)	7 (7.6%)	0 (0.0%)	
	IV	6 (4.2%)	6 (6.5%)	0 (0.0%)	
Operation					
	Wedge	41 (28.9%)	19 (20.7%)	22 (44.0%)	<0.01
	Segmentectomy	13 (9.2%)	7 (7.6%)	6(12.0%)	
	Lobectomy	86 (60.6%)	64 (69.6%)	22 (44.0%)	
	Pneumonectomy	2 (1.4%)	2 (2.2%)	0 (0.0%)	
Histology					
	Whole tumor size (cm)	2.4 ± 1.5	2.1 ± 0.9	2.7 ± 1.8	0.03
	Invasion size (cm)	2.1 ± 1.6	1.1 ± 1.0	2.7 ± 1.6	<0.01
	Invasion size ≥ 2 cm	60 (42.3%)	5 (12.8%)	63 (37.5%)	<0.01
	Non- or nini-invasive AD	17 (12.0%)	0 (18.5%)	17 (0.0%)	<0.01
	Invasive AD	81 (57.0%)	49 (64.1%)	32 (76.0%)	
	SQ	26 (19.2%)	26 (0.0%)	0 (52.0%)	
	ADSQ	1 (0.1%)	1 (0.0%)	1 (2.0%)	
	Others	17 (12.0%)	17 (0.0%)	0 (34.0%)	
	Dominant legion of inv-AD				
	Lepidic	32 (32.7%)	0 (0.0%)	32 (65.3%)	<0.01
	Acinar	39 (39.8%)	32 (65.3%)	7 (14.3%)	
	Papillary	23 (24.0%)	14 (28.5%)	9 (18.4%)	
	Solid	4 (4.1%)	3 (6.1%)	1 (2.0%)	
	V (+)	65 (45.8%)	56 (60.9%)	9 (18.0%)	<0.01
	Ly (+)	78 (54.9%)	64 (69.6%)	14 (28.0%)	<0.01
	Pl (+)	45 (32.6%)	37 (40.2%)	8 (16.0%)	<0.01
	STAS	37 (26.1%)	33 (35.9%)	4 (8.0%)	<0.01
Adjuvant treatment		30 (21.1%)	26 (28.2%)	4 (8.9%)	<0.01

Data are reported as number (%) or mean ± standard deviation. CT, computed tomography; GGO, ground glass opacity; AD, adenocarcinoma; ADSQ, adenosquamous cell carcinoma; CEA, carcinoembryonic antigen; CYFRA, cytokeratin 19 fragment; Ly, lymphatic invasion; min, minimum; Pl, pleural invasion; SQ, squamous cell carcinoma; STAS, spread through air space; SUV, standard uptake value; V, tumor vessel invasion.

**Table 2 cancers-14-04514-t002:** Status of circulating tumor cells according to solid patterns on computed tomography.

	CTC Morphology	Total	Solid	Mixed GGO	*p*-Value
N		142 (100%)	92 (64.8%)	50 (35.2%)	
CTC count					
	All	1.7 ± 3.7	2.2 ± 3.9	0.8 ± 3.0	0.02
	Cluster	0.6 ± 2.0	1.0 ± 2.5	0.0 ± 0.1	<0.01
CTC morphology					
	Cluster	31 (21.8%)	30 (32.6%)	1 (2.0%)	<0.01
	Single	14 (9.9%)	8 (8.7%)	6 (12.0%)	
	Not detected	97 (68.3%)	54 (58.7%)	43 (86.0%)	

Data are reported as number (%) or mean ± standard deviation. CT, computed tomography; GGO, ground glass opacity; CTCs, circulating tumor cells.

**Table 3 cancers-14-04514-t003:** Logistic regression analysis of detecting clustered circulating tumor cells.

Variables	Univariate			Multivariate		
	OR	95% CI	*p*-Value	OR	95% CI	*p*-Value
Pure solid	23.70	3.12–180.00	<0.01	10.30	1.21–87.40	<0.05
Invasive size > 2.0 cm	3.23	1.41–7.41	<0.01	1.30	0.45–3.80	0.6
V (+)	4.72	1.94–11.50	<0.01	1.65	0.48–5.73	0.4
STAS	3.79	1.63–3.83	<0.01	1.06	0.29–3.88	0.9
CEA > 7.5 μg/mL	7.77	3.18–19.00	<0.01	4.33	1.61–11.70	<0.01
CYFRA > 3.3 ng/ml	2.21	0.78–6.23	0.1			
SUVmax > 2.9	9.17	2.63–31.90	<0.01	2.34	0.54–11.30	0.3
Sublobar resection	1.56	0.64–3.81	0.3			
P-stage III, IV	1.08	0.28–4.20	0.9			

OR, odds ratio; CI, confidence interval; V, tumor vessel invasion; STAS, spread through air space; CEA, carcinoembryonic antigen; CYFRA, cytokeratin 19 fragment; SUV, standard uptake value; P, pathological.

**Table 4 cancers-14-04514-t004:** Cox proportional hazard model analyses.

Variables	Univariate			Multivariate		
	HR	95% CI	*p*-Value	HR	95% CI	*p*-Value
RFS						
Sex (Female)	1.27	0.68–2.44	0.5			
Pure solid	5.51	1.68–18.03	<0.01	2.18	0.56–8.39	0.3
Clustered CTC	4.18	2.10–8.29	<0.01	2.48	1.15–5.33	<0.03
Invasive size > 2.0 cm	2.39	1.18–4.84	<0.02	1.26	0.58–2.74	0.6
V (+)	3.03	1.14–6.50	<0.01	1.30	0.50–2.44	0.6
STAS	1.79	0.90–3.56	0.1			
CEA > 7.5 μg/mL	2.20	1.10–4.42	<0.03	1.11	0.50–2.44	0.9
CYFRA > 3.3 ng/ml	1.67	0.68–4.04	0.3			
SUVmax > 2.9	3.23	1.33–7.80	<0.01	1.49	0.53–4.15	0.5
Sublober resection	1.36	0.61–3.02	0.4			
P-stage III, IV	4.56	2.11–9.86	<0.01	3.21	1.42–7.26	<0.01
Adjuvant treatment	0.83	0.40–1.99	0.8			
OS						
Sex (Female)	0.40	0.13–1.25	0.1			
Pure solid	3.29	0.75–14.47	0.1			
Clustered CTC	3.76	1.41–10.06	<0.01	2.66	0.82–8.63	0.1
Invasive size > 2.0 cm	1.63	0.60–4.39	0.3			
V (+)	1.63	0.60–5.94	0.3			
STAS	1.55	0.56–4.26	0.4			
CEA > 7.5 μg/mL	2.97	1.11–8.06	<0.03	2.82	0.86–9.23	0.1
CYFRA > 3.3 ng/ml	5.43	1.91–15.42	<0.01	6.14	2.00–18.80	<0.01
SUVmax > 2.9	1.84	0.59–5.71	0.3			
Sublober resection	0.33	0.12–0.90	<0.03	0.21	0.07–0.63	<0.01
P-stage III, IV	2.87	0.92–8.90	0.07			
Adjuvant treatment	0.40	0.09–1.75	0.2			

HR, hazard ratio; CI, confidence interval; RFS, recurrence free survival; CTC, circulating tumor cell; V, tumor vessel invasion; STAS, spread through air space; CEA, carcinoembryonic antigen; CYFRA, cytokeratin 19 fragment; SUV, standard uptake value; P, pathological; OS, overall survival.

## Data Availability

The data presented in this study are available on request from the corresponding author.

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
