# Peer review of "Pure Solid Pattern of Non-Small Cell Lung Cancer and Clustered Circulating Tumor Cells"

_cancers, 2022, doi:10.3390/cancers14184514_

Round 1

Reviewer 1 Report

In the present manuscript Kawaguchi and colleagues investigated the potential relationship between different patterns detected by computed tomography (CT) and the presence of pre-operative circulating tumor cells (CTCs) in a selected cohort of 142 surgical resected NSCLC  patients.  CT results were compared with the main clinic-pathological data and the CTCs data, collected, and analyzed using a size selection method. Overall, they showed that the pure solid pattern is an independent predictor of the presence of clustered CTSs (≥4 tumor cells) which, in turn, are an independent predictor of poor-recurrence-free survival.

Although preliminary, the results are interesting. The manuscript is clear and well-written.

Some points deserve to be better explained or investigated

1)     The authors choose a size selection method to identify the CTCs. However , there are other different methods to analyse CTCs (for example, the FDA-approved Cell Search system). What is the reason of their choice ?  They used ≥4 tumor cells as the cut-off for clustered CTCs in respect to single cells. Could be the samples with clustered cells furtherly divided according to the number of the clusters ? Since the exact evaluation of CTCs is basic for this study, it would be important to specify and detailed better the analysis. A picture regarding cluster and non-cluster cells could be helpful.

2) Did the authors  correlate the CTC data with any biological and/or molecular features of the 142 surgical resected NSCLC neoplasia?  In particular , do they check any association with EMT characteristics of the neoplasia ?

Author Response

I appreciate your constructive comments. I will response to them point by point.

1)-1 The authors choose a size selection method to identify the CTCs. However, there are other different methods to analyze CTCs (for example, the FDA-approved CellSearch system). What is the reason of their choice?

Thank you for your comment.

CellSearch system is a positive selection method targeting an surface protein: EpCAM. Since this method is a method of hoisting EpCAM positive cells with magnetic antibodies, the sensitivity to c lustered CTC is low, and the larger the cluster, the harder it is to identify. On the other hand, the Size selection method using a filter has high sensitivity to large cluster CTCs, so we think that it is suitable for research targeting Cluster CTCs.

1)-2  They used ≥4 tumor cells as the cut-off for clustered CTCs in respect to single cells. Could be the samples with clustered cells furtherly divided according to the number of the clusters? Since the exact evaluation of CTCs is basic for this study, it would be important to specify and detailed better the analysis. A picture regarding cluster and non-cluster cells could be helpful.

Thank you for your comments.

Representative picture of clustered and singular CTCs is added as Figure 1.

  1. Did the authors correlate the CTC data with any biological and/or molecular features of the 142 surgical resected NSCLC neoplasia? In particular, do they check any association with EMT characteristics of the neoplasia?

Thank you for your comments.

Unfortunately, we do not have information about EMTs for tumor lesions. I understand that this is very interesting and important. We are planning a prospective study in the future, so in this research we will definitely investigate the items you pointed out.

Reviewer 2 Report

In the current study, the authors aimed to evaluate the imlications of a solid pattern on CT and the presence of preoperative clustered CTCs in surgically resected NSCLC patients.

Along with the significant results of the current study, others have demonstrated that progression free and overall survival are significantly correlated with gender, performance status and cancer cell biomarkers (eg CK19, etc) in various NSCLC stages.

Did the authors demonstrated a correlation with any other clinicopathological/epidemiological characteristics?

At least in half (if not in all) patients the heterogeneity of CTCs must be evaluated. Such an evaluation might be performed either at cellular level (immunofluorescence with epithelial and EMT biomarkers) or  molecular level (ngs sequencing). That would better explain which cancer cell characteristics might be responsible for such a worse prognosis.

Author Response

I appreciate your constructive comments. I will response to them point by point.

  1. Along with the significant results of the current study, others have demonstrated that progression free and overall survival are significantly correlated with gender, performance status and cancer cell biomarkers (eg CK19, etc) in various NSCLC stages.

Thank you for your comment. CYFRA and sex difference are added to the analyses of prognosis. Since the patients assessed were surgical candidate, there was little difference in performance status, which has not chosen as a variable.

Although CYFRA is an independent prognostic indicator of OS, this result didn’t affect the interpretation of our study.

  1. Did the authors demonstrated a correlation with any other clinicopathological/epidemiological characteristics?

Thank you for your comment.

Unfortunately, no more clinicopathological/epidemiological characteristics were assessed. If important factors are pointed out, I think it will be reflected in the prospective research we are planning in the future.

  1. At least in half (if not in all) patients the heterogeneity of CTCs must be evaluated. Such an evaluation might be performed either at cellular level (immunofluorescence with epithelial and EMT biomarkers) or molecular level (ngs sequencing). That would better explain which cancer cell characteristics might be responsible for such a worse prognosis.

Thank you for your constructive comment.

As you mention, it is very important to clarify the cytological characteristics of CTC, especially the status of EMT. Unfortunately, this study did not allow us to even look at the morphological characteristics of CTC. This is due to limitations of the extraction method by ScreenCell. Currently, we are developing a CTC extraction filter that can easily immunostainize and are proceeding with the procedure for practical application, so we will try to consider it in more detail referring to your points in future research.

Round 2

Reviewer 1 Report

The authors have satisfactorily answered to the questions. I just recommend the authors to introduce the comments into the appropriate section of the  text.

Author Response

I appreciate your comment. The comments I responded were introduced into the appropriate section of the text where is highlighted.

Reviewer 2 Report

The authors have replied in all points addressed.

The manuscript is suggested to be accepted in the present form

Author Response

I appreciate your comments.